# Germline pathogenic variants of 11 breast cancer genes in 7,051 Japanese patients and 11,241 controls

Yukihide Momozawa[1], Yusuke Iwasaki[1], Michael T. Parsons[2], Yoichiro Kamatani[3], Atsushi Takahashi[3,4], Chieko Tamura[5], Toyomasa Katagiri[6], Teruhiko Yoshida[7], Seigo Nakamura[8], Kokichi Sugano[7,9], Yoshio Miki[10], Makoto Hirata[7,11], Koichi Matsuda[12], Amanda B. Spurdle[2] & Michiaki Kubo[1]

Pathogenic variants in highly penetrant genes are useful for the diagnosis, therapy, and surveillance for hereditary breast cancer. Large-scale studies are needed to inform future testing and variant classification processes in Japanese. We performed a case-control association study for variants in coding regions of 11 hereditary breast cancer genes in 7051 unselected breast cancer patients and 11,241 female controls of Japanese ancestry. Here, we identify 244 germline pathogenic variants. Pathogenic variants are found in 5.7% of patients, ranging from 15% in women diagnosed <40 years to 3.2% in patients ≥80 years, with BRCA1/2, explaining two-thirds of pathogenic variants identified at all ages. BRCA1/2, PALB2, and TP53 are significant causative genes. Patients with pathogenic variants in BRCA1/2 or PTEN have significantly younger age at diagnosis. In conclusion, BRCA1/2, PALB2, and TP53 are the major hereditary breast cancer genes, irrespective of age at diagnosis, in Japanese women.

[1] Laboratory for Genotyping Development, RIKEN Center for Integrative Medical Sciences, 1-7-22 Suehiro-cho, Tsurumi-ku, Yokohama City, Kanagawa 230-0045, Japan. [2] Division of Genetics and Population Health, QIMR Berghofer Medical Research Institute, 300 Herston Rd, Herston, Brisbane, QLD 4006, Australia. [3] Laboratory for Statistical Analysis, RIKEN Center for Integrative Medical Sciences, 1-7-22 Suehiro-cho, Tsurumi-ku, Yokohama City, Kanagawa 230-0045, Japan. [4] Department of Genomic Medicine, Research Institute, National Cerebral and Cardiovascular Center, 5-7-1 Fujishiro-dai, Suita, Osaka 565-8565, Japan. [5] FMC Tokyo Clinic, 1-3-2, Iidabashi, Chiyoda-ku, Tokyo 102-0072, Japan. [6] Division of Genome Medicine, Institute for Genome Research, Tokushima University, 3-18-15 Kuramoto, Tokushima 770-8503, Japan. [7] Department of Genetic Medicine and Services, National Cancer Centre Hospital, 5-1-1 Tsukiji, Chuo-ku, Tokyo 104-0045, Japan. [8] Division of Breast Surgical Oncology, Department of Surgery, Showa University School of Medicine, 1-5-8 Hatanodai, Shinagawa-ku, Tokyo 142-8666, Japan. [9] Oncogene Research Unit/Cancer Prevention Unit, Tochigi Cancer Centre Research Institute, 4-9-13 Yohnan, Tochigi 320-0834, Japan. [10] Department of Molecular Genetics, Medical Research Institute, Tokyo Medical and Dental University, 1-5-45 Yushima, Bunkyo-ku, Tokyo 113-8510, Japan. [11] Laboratory of Genome Technology, Human Genome Center, Institute of Medical Science, The University of Tokyo, 4-6-1 Shirokanedai, Minato-ku, Tokyo 108-8639, Japan. [12] Graduate School of Frontier Sciences, The University of Tokyo, 4-6-1 Shirokanedai, Minato-ku, Tokyo 108-8639, Japan. Correspondence and requests for materials should be addressed to Y.M. (email: momozawa@riken.jp) or to M.K. (email: michiaki.kubo@riken.jp)

Breast cancer is the most common cancer in women worldwide[1]. Although several factors such as age, reproductive history, and oral contraceptives are known to contribute to the development of breast cancer, genetic factors also play an important role[2]. Pathogenic variants in highly penetrant hereditary breast cancer genes, such as *BRCA1* and *BRCA2* are known to account for 5–10% of breast cancer in the general population[3,4]. Association between protein-truncating variants in 11 different genes and breast cancer risk has been established[3]. Although, the precise risk of each gene was uncertain, sequencing of these genes have been recommended to provide personalized diagnosis, therapy, and surveillance for the high-risk patients and their relatives[5].

Clinical sequencing using multi-gene panel testing has been widely used for genetic testing of various diseases, including hereditary breast cancer[3]. However, this multi-gene panel testing detects many variants of uncertain clinical significance[6]. Uncertain classification or misclassification[7] of variants can be partially solved by filtering the variants through the allele frequency database of various populations, such as the Exome Aggregation Consortium (ExAC)[8]. For variants with lower allele frequencies, additional clinical information including segregation or large-scale case-control analysis is needed to resolve variant classification[8]. In addition, since almost all the available data are from the population of the European descendent, it is unclear whether clinical interpretation are generally applicable to other populations[3].

Here we sequenced 11 established hereditary breast cancer genes[3] in 7051 unselected women with breast cancer and 11,241 controls to estimate the contribution of germline pathogenic variants in these genes to breast cancer in the Japanese population. We also compared the clinical characteristics of breast cancer patients with versus without a germline pathogenic variant. We identify 244 germline pathogenic variants and we show demographic and clinical characteristics of patients with pathogenic variants. We conclude that *BRCA1/2*, *PALB2*, and *TP53* are the major hereditary breast cancer genes, irrespective of age at diagnosis, in Japanese women.

## Results

**Patient characteristics**. Characteristics of female study patients were shown in Table 1. Mean age at diagnosis of breast cancer was 55.8 years old. Among them, 0.7% had prior or concurrent ovarian cancer. Family history of cancer types known to be associated with hereditary breast cancer syndromes was reported by breast cancer cases as follows: 11.8% breast, 1.2% ovary, 3.5% pancreas, 2.9% prostate, and 0.8% thyroid cancer. Characteristics of male patients were shown in Supplementary Table 1.

**Pathogenic germline variants in women**. Sequencing of the 11 established hereditary breast cancer genes identified 1781 germline variants among 7051 breast cancer cases and 11,241 controls (Supplementary Data 1). We annotated clinical significance of each variant using the association results, known clinical significance in ClinVar, population data, computational data, and functional data following the the American College of Medical Genetics and Genomics and the Association for Molecular Pathology (ACMG/AMP) guidelines. After comparing with ClinVar, we classified 244 variants as pathogenic, 356 as benign and 1181 as VUS (Supplementary Note 1). Among VUS, two had conflicting evidence for pathogenic and benign criteria, while the others did not have enough evidence. Among 244 pathogenic variants, 204 were disruptive, 38 were non-synonymous, and 2 (p. Gln1395Gln in *BRCA1*[9] and p.Pro3039Pro in *BRCA2*[10]) were synonymous but reported to alter splicing, respectively. Among

### Table 1 Characteristics of study population in women

| Variable | | Breast cancer patients (%) | Controls (%) * |
|---|---|---|---|
| No. of subjects | | 7093 | 11,260 |
| Age at entry (Mean ± SD) | Years old | 59.1 ± 12.0 | 72.0 ± 7.5 |
| Age at diagnosis (Mean ± SD) | Years old | 55.8 ± 12.0 | — |
| Personal history of ovarian cancer# | Yes (%) | 47 (0.7) | 0 (0.0) |
| | No | 7046 | 11,260 |
| Family history of breast cancer | Yes (%) | 838 (11.8) | 0 (0.0) |
| | No | 6255 | 11,260 |
| Family history of ovarian cancer | Yes (%) | 83 (1.2) | 0 (0.0) |
| | No | 7010 | 11,260 |
| Family history of pancreas cancer | Yes (%) | 247 (3.5) | 0 (0.0) |
| | No | 6846 | 11,260 |
| Family history of prostate cancer | Yes (%) | 207 (2.9) | 0 (0.0) |
| | No | 6886 | 11,260 |
| Family history of thyroid cancer | Yes (%) | 54 (0.8) | 0 (0.0) |
| | No | 7039 | 11,260 |

Family history of cancer refers to reported cancer in first and/or second-degree relative
*Controls without past history nor family history of cancers were selected for this study
#Personal history of ovarian cancer includes prior or concurrent ovarian cancer

the 244 pathogenic variants, 131 (53.9%) variants were newly identified in this study, with the proportion of novel pathogenic variants ranging from 100% (*STK11*, 1 unique variant only; *NBN*, 3 unique variants) to 25% (*BRCA1*, 55 variants total) (Supplementary Fig. 1). The proportion of pathogenic variants among all variants differed by gene from 1.3% for *STK11* to 33.3% for *PTEN*, while the proportion of benign variants ranged from 11.1% for *PTEN* to 27.0% for *CDH1* (Supplementary Table 2).

Supplementary Fig. 2 shows the location of pathogenic variants. Most variants (75.8%) were singletons and 11.0% were doubletons. We identified 15 frequent pathogenic variants found in 5 or more patients in *ATM* (p.Ile2629fs), *BRCA1* (p.Leu63*, p. Gln934*, p.Lys1095Glu, and p.Tyr1874Cys), *BRCA2* (p.Ile605fs, p.Ile1859fs, p.Ser1882*, p.Asn2135fs, p.Arg2318*, p.Gln3026*, and p.Pro3039Pro), *CHEK2* (p.Ala523Thr), *PALB2* (p.Gln559*), and *TP53* (p.Arg248Gln) in Fig. 1. Four variants in *BRCA1* (p. Lys1095Glu and p.Tyr1874Cys), *CHEK2* (p.Ala523Thr), and *PALB2* (p.Gln559*) were novel pathogenic variants in this study.

In total, pathogenic variants were found in 404 (5.7%) breast cancer cases and 67 (0.6%) controls (Fisher's exact test, $P = 2.87 \times 10^{-102}$, odds ratio (OR) = 10.1). When we performed a gene-based test using pathogenic variants, four genes were significantly associated with breast cancer (*BRCA2*: $P = 9.87 \times 10^{-58}$, OR = 16.4; *BRCA1*: $P = 3.71 \times 10^{-36}$, OR = 33.0; *PALB2*: $P = 5.79 \times 10^{-8}$, OR = 9.0; and *TP53*: $P = 5.93 \times 10^{-5}$, OR = 8.5, Table 2). In addition, *PTEN*, *CHEK2*, *NF1*, and *ATM* showed nominal association ($P < 0.05$). No association was observed for pathogenic variants in *CDH1* (2 cases), *NBN* (1 case, 3 controls), and *STK11* (1 control). The associated eight genes explained 99.3% of patients with pathogenic variants in this study (Supplementary Fig. 3A). We also found four breast cancer cases that had two pathogenic variants (Supplementary Table 3). One patient had two pathogenic 1-bp deletions (p.Ile917fs and p. Lys918fs) in *BRCA1*. Since sequencing analysis showed two variants cause frame-shift only in one allele, this patient was considered as a single carrier with a pathogenic variant in *BRCA1* for further study. The other three patients had two pathogenic variants in different genes (a *BRCA2* truncating variant plus a pathogenic variant in *ATM* or *CHEK2*), and the number of double carriers were significantly smaller than expected if we

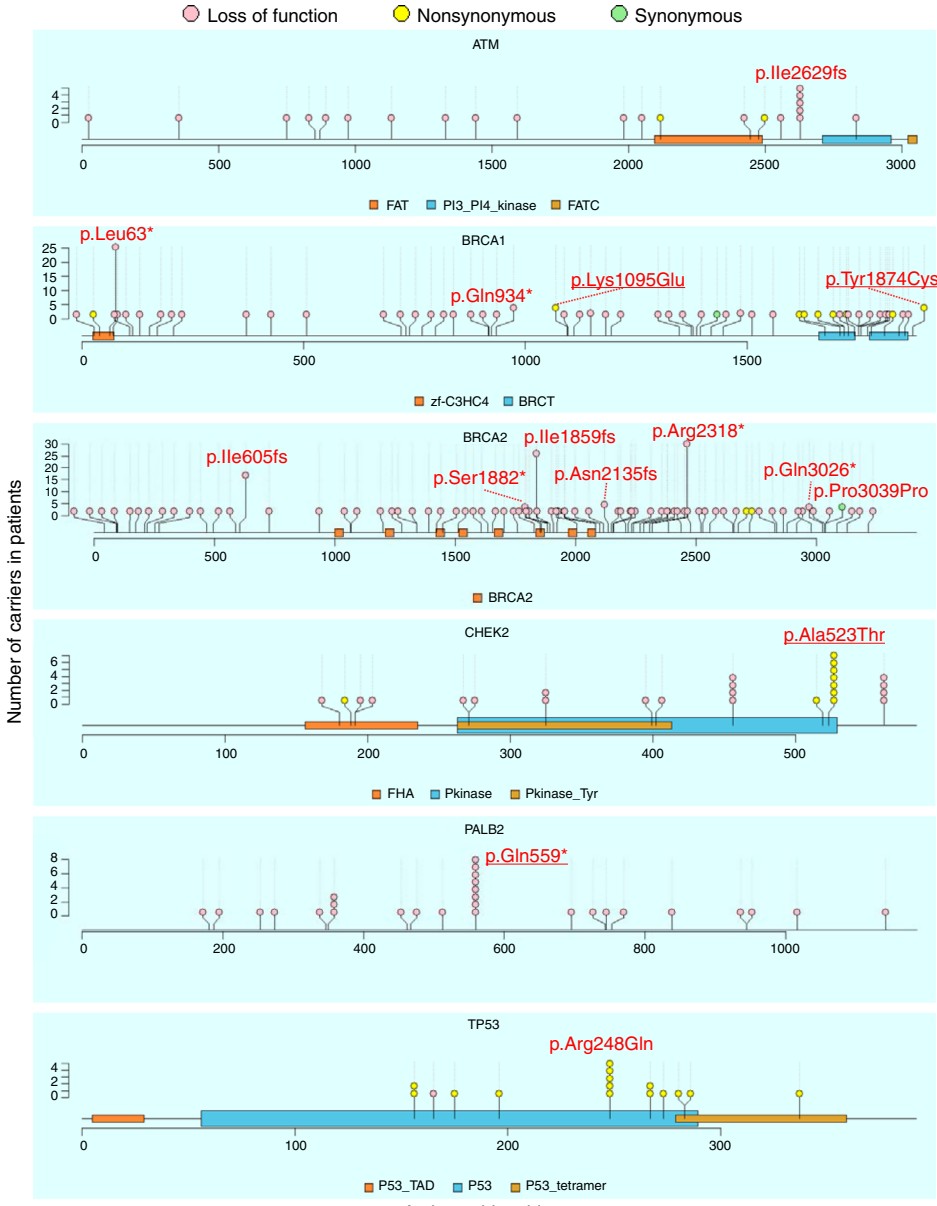

**Fig. 1** Location and the number of frequent pathogenic variants in six genes in Japanese breast cancer women. Locations of frequent pathogenic variants found in patients and domains in proteins are shown by lollipop structures, with the variant type indicated by color. Pink, yellow, and green circles indicates loss of function, non-synonymous, and synonymous variants, respectively. The x-axis reflects the number of amino acid residues, and the y-axis shows the total number of patients with each pathogenic variant. HGVS.p of frequent variants with five or more patients are shown and four variants newly identified as pathogenic variants are underlined

hypothesized that pathogenic variants were randomly distributed to each patient ($P = 2.75 \times 10^{-3}$). Clinical characteristics of the three double carriers are shown in Supplementary Table 3.

**Impact of pathogenic variants on clinical characteristics**. To investigate the impact of pathogenic variants on clinical characteristics of breast cancer, we compared clinical characteristics between the patients with pathogenic variants and those without any pathogenic variant (Supplementary Table 4). Breast cancer patients with pathogenic variants had significantly younger age at diagnosis, higher frequencies of ovarian cancer, bilateral breast cancer, advanced clinical stage, triple negative breast cancer and family history of breast, ovary, pancreas, gastric, liver, bone, and bladder cancer, respectively.

We further examined the impact of pathogenic variants on the age at diagnosis of breast cancer by stratifying age at diagnosis into 10-year age groupings (Fig. 2a). Pathogenic variants were found in 15.0% of patients diagnosed at less than 40 years old. The proportion of pathogenic variants significantly decreased with advancing age at diagnosis (Cochran-Armitage test, $P = 1.50 \times 10^{-15}$). However, we still observed pathogenic variants in 3.2% of breast cancer patients diagnosed at 80 years old or over. When we examined the age at diagnosis of breast cancer by gene, we found that the patients with pathogenic variants in *PTEN*, *BRCA1*, and *BRCA2* were significantly younger at diagnosis compared to patients without pathogenic variants (Table 3). Indeed, 8 of 11 patients with pathogenic variants in *PTEN* were diagnosed at <40 years of age and *PTEN* alterations were the third most common (after *BRCA1* and *BRCA2*) in patients <40 years

**Table 2 Result of gene-based association test using pathogenic variants**

| Gene | No. of pathogenic variants | Case (n = 7051) | Control (n = 11,241) | P-value* | OR | (95% CI) |
|---|---|---|---|---|---|---|
| | | No. of carriers (%) | No. of carriers (%) | | | |
| BRCA2 | 85 | 191 (2.71) | 19 (0.17) | $9.87 \times 10^{-58}$ | 16.4 | (10.2–28.0) |
| BRCA1 | 55 | 102 (1.45) | 5 (0.04) | $3.71 \times 10^{-36}$ | 33.0 | (13.7–103.8) |
| PALB2 | 21 | 28 (0.40) | 5 (0.04) | $5.79 \times 10^{-8}$ | 9.0 | (3.4–29.7) |
| TP53 | 13 | 16 (0.23) | 3 (0.03) | $5.93 \times 10^{-5}$ | 8.5 | (2.4–45.6) |
| PTEN | 12 | 11 (0.16) | 1 (0.01) | $2.16 \times 10^{-4}$ | 17.6 | (2.6–753.3) |
| CHEK2 | 17 | 26 (0.37) | 13 (0.12) | $4.31 \times 10^{-4}$ | 3.2 | (1.6–6.8) |
| NF1 | 8 | 8 (0.11) | 0 (0.00) | $4.86 \times 10^{-4}$ | Inf | (2.7–Inf) |
| ATM | 27 | 22 (0.31) | 17 (0.15) | 0.031 | 2.1 | (1.0–4.1) |
| CDH1 | 2 | 2 (0.03) | 0 (0.00) | 0.149 | Inf | (0.3–Inf) |
| NBN | 3 | 1 (0.01) | 3 (0.03) | 1.000 | 0.5 | (0.0–6.6) |
| STK11 | 1 | 0 (0.00) | 1 (0.01) | 1.000 | 0.0 | (0.0–62.1) |
| Sum | 244 | 404# (5.73) | 67 (0.60) | $2.87 \times 10^{-102}$ | 10.1 | (7.8–13.4) |

*Fisher's exact test
#Sum of carriers from the 11 genes were 407. However, three patients had two pathogenic variants in different genes. Thus, the number of carriers became 404

old (Fig. 2b). However, when we divided patients into two groups by 50 years of age at diagnosis according to the definition of the National Comprehensive Cancer Network (NCCN) guidelines[5], the proportion of causative genes was not different between the early-onset and late-onset of breast cancer ($\chi^2$-test, $P = 0.155$, Supplementary Fig. 3B, C).

**Male breast cancer**. We conducted the same analysis in 53 unselected male cases and 12,490 controls. We identified 75 pathogenic variants (Supplementary Data 2) in 13 of 53 (24.5%) male breast cancer patients and 129 of 12,490 (1.0%) male controls (Fisher's exact test, $P = 1.64 \times 10^{-14}$, OR = 31.1, Supplementary Table 5). One patient had two pathogenic variants (p.Thr3033fs in BRCA2, and p.Val475Met in CDH1). Compared to female breast cancer patients, the frequency of pathogenic variants in male breast cancer patients was significantly higher (Fisher's exact test, $P = 7.93 \times 10^{-6}$, OR = 5.3). The frequencies of pathogenic variants in male controls were also higher than female controls (1.0% in male and 0.6% in female, $P = 2.31 \times 10^{-4}$). When we performed a gene-based test, BRCA2 was significantly associated with male breast cancer (18.9% in cases and 0.2% in controls, Fisher's exact test, $P = 1.73 \times 10^{-16}$, OR = 111.2). All pathogenic variants were found in a single breast cancer patient (Supplementary Fig. 4). CDH1 and BRCA1 were nominally associated ($P < 0.05$) although only one or two patients had a pathogenic variant in these genes (Supplementary Table 5). When age at diagnosis of breast cancer was compared between male breast cancer patients with pathogenic variants in BRCA2 and those with no pathogenic variant, age at diagnosis of breast cancer was significantly older in patients with pathogenic variants (mean ± SD, 75.5 ± 5.8 years old) than those with no pathogenic variant (63.3 ± 10.6 years old, t-test, $P = 3.90 \times 10^{-5}$).

**Discussion**
We identified 1781 germline variants in the 11 established hereditary breast cancer genes in 7051 breast cancer patients and 11,241 controls in women. Although ClinVar has registered many pathogenic variants of 11 genes, more than half of the 244 pathogenic variants were newly identified in this study. Pathogenic variants were found in 5.7% of unselected Japanese breast cancer patients. BRCA1, BRCA2, PALB2, and TP53 were the significant causative genes. Proportion of pathogenic variants was high in younger age at diagnosis and gradually decreased with advancing age at diagnosis. However, we still found the patients with pathogenic variants diagnosis in elderly women. In addition

to BRCA1/2, we found pathogenic variants in PTEN are associated with younger age at onset of breast cancer in Japanese women.

The 11 genes analyzed in this study have been reported previously as hereditary breast cancer genes, but the strength of evidence for association of each gene with breast cancer and disease risk varies. Further, published risk estimates are likely to be inflated for at least some genes due to ascertainment bias[3]. We observed a significant contribution to breast cancer risk in BRCA1/2, PALB2, and TP53. The disease risks of BRCA1/2 and PALB2 are comparable to that previously reported[3], but the risk of TP53 is largely different (8.5 in this study and 105 in the previous meta-analysis[3]). This is likely explained by several factors. Firstly, previous estimates were based on studies of familial patients presenting with clinical features of Li–Fraumeni syndrome, whereas in this study we calculated disease risk for women unselected for family history of cancer. Second, functional effects differ between variants in TP53, which causes a wide range of symptoms, from the severe form known as Li–Fraumeni to the less severe non-syndromic predisposition[11], and it is possible that the variants found in patients with unselected breast cancer have less impact on protein function than those identified in patients with classical Li–Fraumeni syndrome. Among four other genes showing $P < 0.05$ (PTEN, CHEK2, NF1, and ATM), the disease risks of ATM and CHEK2 were comparable to previous reports[3]. Disease risks for PTEN and NF1 were not reliably estimated, despite strong evidence for association ($P < 5 \times 10^{-4}$), due the low numbers of carriers, indicating need for even larger studies to estimate risk at the population level. Although the association with breast cancer for CDH1 and STK11 has been reported previously for patients for hereditary diffuse gastric cancer[12] and Peutz–Jeghers syndrome[13], only two and zero Japanese breast cancer patients, respectively, had a pathogenic variant in these genes. That is, CDH1 and STK11 have a limited contribution to breast cancer in unselected Japanese women. The reported contribution of NBN to breast cancer risk was mainly based on one-specific variant (c.657del5, rs587776650) in the Slavic population[3,14], which was not observed in the Japanese population. Other NBN variants designated as pathogenic using ACMG/ AMP criteria were observed in only 1 case and 3 controls, providing little support for a role of NBN in Japanese unselected breast cancer patients. However, our study has confirmed the importance of the remaining eight genes in genetic testing in Japan and jointly assessed the disease risk of each gene.

A recent study reported the proportion of pathogenic variants was 9.3% among 35,409 multi-ethnic women with a single

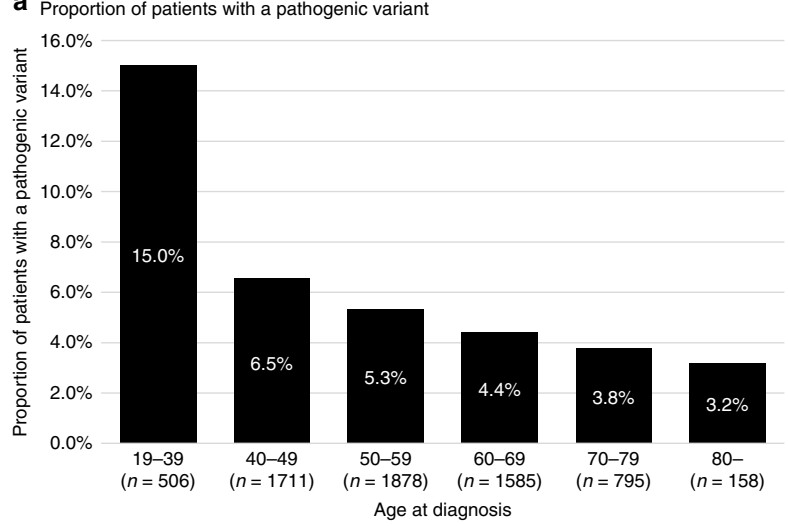

**a** Proportion of patients with a pathogenic variant

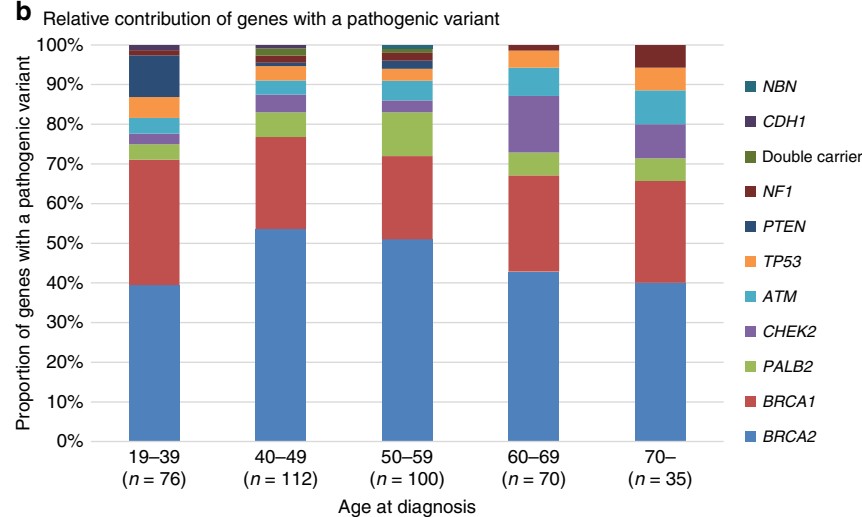

**b** Relative contribution of genes with a pathogenic variant

**Fig. 2 a** Proportion of patients with pathogenic variants and **b** relative contribution of genes by the age at diagnosis of breast cancer women in 10-year-age groupings. **a** Proportion of patients with a pathogenic variant significantly decreased with advancing age (Cochran-Armitage test, $P = 1.50 \times 10^{-15}$). **b** Color indicates each gene as shown in the right legend

**Table 3 Mean age at diagnosis of breast cancer in patients with pathogenic variants**

| Gene with pathogenic variant | Number of patients* | Mean ± SD | P-value[#] |
|---|---|---|---|
| No pathogenic variants | 6240 | 56.1 ± 11.9 | Reference |
| BRCA2 | 185 | 51.0 ± 11.5 | $9.47 \times 10^{-9}$[†] |
| PTEN | 11 | 36.6 ± 10.5 | $1.04 \times 10^{-4}$[†] |
| BRCA1 | 97 | 50.9 ± 13.0 | $1.61 \times 10^{-4}$[†] |
| Double carrier | 3 | 48.3 ± 6.7 | 0.180 |
| TP53 | 16 | 50.6 ± 16.3 | 0.193 |
| PALB2 | 27 | 52.9 ± 12.7 | 0.194 |
| CDH1 | 2 | 42.0 ± 7.1 | 0.217 |
| CHEK2 | 23 | 57.9 ± 12.7 | 0.514 |
| NF1 | 8 | 59.5 ± 17.8 | 0.607 |
| ATM | 20 | 54.7 ± 14.1 | 0.659 |
| NBN | 1 | 56.0 | — |

*The number of patients with age at diagnosis is shown
[#]Mean age at diagnosis of each gene was compared with the patients without pathogenic variants by t-test
[†]Significant after the Bonferroni correction was applied

diagnosis of breast cancer who underwent clinical genetic testing with a 25-gene panel of hereditary cancer genes[15]. When we selected 3136 patients meeting NCCN guidelines (Supplementary Note 2) and compared the proportion of patients with a pathogenic variant within the 11 genes analyzed in this study, the proportion of breast cancer patients with pathogenic variants was similar (Supplementary Table 6), indicating that these clinical criteria have utility in the Japanese population. However, the carrier frequencies in BRCA2 and PTEN were significantly higher in Japanese, while those in ATM and CHEK2 were lower compared to the multi-ethnic study[15]. One possible explanation might be the differences in ancestry-specific variants. In our study we did not identify the CHEK2 1100delC variant, reported in 0.3–3.8% of patients in Europe, North America, and Oceania[16], while three frequent BRCA2 variants (p.Ile605fs, p.Ile1859fs, and p.Arg2318*) explaining 39.8% of pathogenic BRCA2 variant carriers in our study were not found in 1824 patients of mostly European descent[17]. Therefore, it is important to assess the contribution of pathogenic variants in hereditary breast cancer genes using large number of samples in each population.

We identified 113 variants previously noted as pathogenic. The data from this study helped to classify 131 additional variants,

resulting in a total of 244 pathogenic variants identified. This increase resulted in the identification of 57% more patients (from 258 to 404) with a pathogenic variant. Supplementary Fig. 5 shows this change in each gene. Although more than 75% of patients could be identified by only ClinVar in *BRCA1/2*, only a small proportion of patients with other genes, especially *PALB2* (18%), *CHEK2* (8%), *ATM* (24%), and *NF1* (25%), were identified. Therefore, this study contributes to improved identification of patients with a pathogenic variant, especially in genes other than *BRCA1/2*, in the diagnosis of hereditary breast cancer in clinical practice in Japan. Next we investigated the proportion of pathogenic variants shared between other Asian countries and this study to address how the Japanese data are relevant to other Asian populations. Two studies from China[18] and Malaysia[19] sequenced *BRCA1/2* in >2000 selected and unselected breast cancer patients, respectively. The Chinese study identified 175 unique pathogenic variants in 247 of 2991 (8.3%) patients. Of the 175 pathogenic variants, 15 (8.6%) pathogenic variants were identified in this study. Similarly, the Malaysian study identified 97 unique pathogenic variants in 121 of 2575 (4.7%) patients. Of these pathogenic variants, 15 (15.5%) variants were identified in our study. These results suggest that pathogenic variants identified in this study were shared in Asian populations to some extent. Therefore, this study contributes to the identification of patients with a pathogenic variant in the diagnosis of hereditary breast cancer in other Asian countries. However, it will still be necessary to create a list of pathogenic variants based on a large number of samples for improved diagnosis of hereditary breast cancer.

This study has limitations. First, risk of pathogenic variants might be overestimated, because we did not use unselected individuals from the general population as controls. However, since pathogenic variants were found even in breast cancer patients diagnosed at elderly, small number of subjects with pathogenic variants who will develop breast cancer might be included in the general population. Second, our method could sequence full coding regions with high quality, but could not detect large rearrangements and deletions known to cause hereditary breast cancer. However, the frequency of these rearrangements in pathogenic variants is reported to be low for the *BRCA1/2* genes[20–22].

In conclusion, because all breast cancer patients in this study were collected as unselected breast cancer from all over Japan and a large number of cancer-free controls (Supplementary Note 3, 4) was jointly analyzed by the same method[23], the findings in this study provide important data to guide genetic testing for breast cancer susceptibility genes in Asian population.

## Methods

**Study population**. We obtained all study samples from the Biobank Japan[24,25], which is a multi-institutional hospital based registry that collects DNA from peripheral blood leukocytes and clinical information from patients with various common diseases, including breast cancer, from all over Japan[26]. In this study, we analyzed all 7093 female breast cancer patients with DNA available for sequencing. We also selected 11,260 female controls who were 60 years old or over and do not have past history nor family history of cancers. Clinical characteristics of cases and controls were collected by an interview or medical record survey using a standard questionnaire at the entry to the Biobank Japan. We also examined 53 male breast cancer patients and 12,520 male controls using the same criteria. We analyzed women and men separately, as genetic risk for hereditary breast cancer differs between men and women[27] (Supplementary Note 3).

All individuals who participated in this study provided written inform consent. This study was approved by the ethical committees of the Institute of Medical Sciences, the University of Tokyo and RIKEN Center for Integrative Medical Sciences.

**Sequencing and bioinformatics analysis**. In this study, we analyzed all coding regions and 2 bp flanking intronic sequences of the 11 established genes causing hereditary breast cancer[3]. All transcripts registered in Consensus CDS (CCDS)

release 15[28] for each gene were analyzed (Supplementary Table 7). A total length of the target region was 48,716 bp. A multiplex PCR-based target sequencing method was used to sequence the target region[29]. We used a two-step PCR method to construct DNA libraries. The 1st PCR (25 cycle) was performed with 471 primer pairs and 2X Platinum Multiplex PCR Master Mix (Thermo Fisher Scientific) to amplify the target region, followed by the 2nd PCR (4 cycle) with 8-bp barcode and adapter sequences added using primers targeting shared 5' overhangs introduced during the 1st PCR and KAPA HiFi HotStart DNA Polymerase (KAPA). After purification and quantification of pooled libraries, we sequenced them by $2 \times 150$-bp paired-end reads on a HiSeq 2500 (Illumina) instrument. Sequence reads allocated to each individual were aligned to the human reference sequence (hg19) using Burrows-Wheeler Aligner (ver. 0.7.12)[30] and processed using Genome Analysis Toolkit (GATK, ver. 3.4-46)[31]. For quality control, we selected individuals in which more than 98% of the target region was covered with 20 or more sequencing reads.

We called variants of each individual separately using UnifiedGenotyper and HaplotypeCaller of GATK, and VCMM (ver. 1.0.2)[32]. Genotypes for all individuals were jointly determined for each variant based on the sequencing read ratio of reference and alternative alleles. When the alternative allele frequency was between 0 and 0.15, between 0.25 and 0.75, and between 0.85 and 1, we assigned homozygote of the reference allele, heterozygote, and homozygote of the alternative allele, respectively. We excluded variants with call rates <98% or variants that did not follow Hardy–Weinberg equilibrium in controls ($P < 1 \times 10^{-6}$)[33]. Finally, 99.95% of the target region on average was covered with 20 or more sequence reads in 7051 female cases, 11,241 female controls, 53 male cases, and 12,490 male controls.

**Annotation of variants**. Clinical significance of each variant was annotated according to the ACMG/AMP guidelines[7,34] using association results in this study, known clinical significance information from ClinVar[35], population data from the 1000 genomes project[36], ExAC[8] and Tohoku Medical Megabank Organization (ToMMo)[37], computational data by in silico programs, and functional data (Supplementary Note 1). After the annotation, the results were compared with classifications in ClinVar to identify additional information and determine the final classification of each variant, collapsed from a 5-tier to 3-tier classification system: pathogenic, benign, and uncertain significance. The annotation procedure is detailed in Supplementary Note 1. All these annotations of each variant were initially performed by YMo, YI, and MK, and reviewed by Japanese experts (T.K., T.Y., S.N., K.S., and Y.Mi.). Annotations were further reviewed by M.P. and A.B.S., members of the ClinGen-approved BRCA expert panel set up by the Evidence-based Network for the Interpretation of Germline Mutant Alleles (ENIGMA) consortium[38], to assess the consistency of interpretation of the ACMG/AMP guidelines.

**Statistical analysis**. Case control association analysis was performed by Fisher's exact test under a dominant model. We considered $P = 1 \times 10^{-4}$ as the threshold for gene-based tests, as recommended previously for breast cancer risk assessment[3]. OR and 95% confidence interval (CI) of each variant were also calculated.

To estimate the effect of pathogenic variants on clinical characteristics, we used *t*-test for continuous variables and Fisher's exact test or Cochran-Armitage test for discrete variables. Proportions of predisposition genes in patients with pathogenic variants by age at diagnosis were compared by $\chi^2$-test. $P < 0.05$ was considered statistically significant. Bonferroni correction was applied for multiple comparisons. All analysis was performed with R statistical package (ver. 3.1.3).

## Data availability

Sequence data has been deposited at the Japanese Genotype-phenotype Archive (JGA, http://trace.ddbj.nig.ac.jp/jga), which is hosted by the DDBJ, under accession number JGAS00000000140.

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

### Acknowledgements

We thank the people who participated in this study. We acknowledge C. Inai, N. Hakozaki, H. Hashinokuchi, and other staffs of the Laboratory for Genotyping Development, RIKEN Center for the Integrative Medical Sciences, and the staff of the BioBank Japan project. This work was conducted as part of the BioBank Japan Project supported by the Japan Agency for Medical Research and Development and by the Ministry of Education, Culture, Sports, Sciences, and Technology of the Japanese government. A.B. Spurdle is supported by an Australian NHMRC Senior Research Fellowship.

### Author contributions

Y.Mo. and M.K. conceived and designed the study. Y.Mo. and Y.I. performed experiments and Y.Mo., Y.I., M.P., A.B.S and M.K., were involved with analysis of the data. All other authors reviewed results. Y.Mo., M.P., A.B.S and M.K. drafted the manuscript subsequently redrafting the manuscript with input from all authors. All authors approved the final version of the report.

### Additional information

**Competing interests:** The authors declare no competing interests.

