## [Peer Review File · Nature Communications]

Reviewers' Comments:

Reviewer #1:

Remarks to the Author:

This study records the frequency of germline pathogenic mutations in 11 breast cancer predisposition genes among a Japanese population of 7051 women with breast cancers and 11,241 female controls. In addition, 53 male breast cancer patients and 12,520 male controls were studied. Similar studies have been extensively reported on in Western European populations but this is the largest study of a Japanese population. A strength of the study is the large sample size and the inclusion of controls that seem to be reasonably matched to the cases. Technically the study appears to have been conducted well and the variant annotation and pathogenicity calling has been conducted rigorously.

Overall the study provides some insight into the prevalence of breast cancer predisposition gene mutations in an Asian population.

There are a number of aspects of the manuscript that limit the impact of the work:

1. The extent to which this data is relevant to other Asian populations where there have been few large-scale studies of this kind is unclear. It would be helpful if the authors could indicate how the Japanese population data might reflect Asian populations in general? How will this work change clinical practice in Japan and elsewhere?

2. The male breast cancer data is based on small numbers but it is not unreasonable to include these in the study. However I am confused as to why the male controls are not included in the overall analysis. While there might be expected to be some minor differences in the frequency of pathogenic mutations in males because there will be no attrition of carriers with breast and ovarian cancer, surely this would be relatively minor?

3. From my perspective the aspect of the study that raises it above the numerous similar studies in European populations is how the frequency of mutations and carrier characteristics differ in Japan and to what extent the lessons learned from western populations will be applicable in Japan and other Asian populations. While these aspects are noted in the discussion, much of the manuscript is devoted to a description of the variant classification, VUS, number of novel variants, distribution of variants across each of the genes and relationship with gene length etc. While this data might should be included as supplementary data, too much of the results and discussion are devoted to these aspects. Overall the manuscript could be shortened considerably by focussing on the novel aspects.

4. On page 5 the authors state that they studied 11 established hereditary breast cancer genes but in presenting the data the authors state that only some of these reached significance. Given that most of these genes have very well established roles in breast cancer predisposition, why are those other genes de-emphasised. Particularly for STK11, TP53 and CDH1 where the number of carriers is small, these will never reach significance but they are well established cancer predisposition genes. On the other hand, genes such as NBN are not well supported in the literature and the data presented here shows only one carrier in the cases and 3 in the controls. This result should be noted and compared with previous studies which I believe also show no association with breast cancer predisposition.

5. Some of the comments in the manuscript describe features of breast cancer predisposition that have been established for many years in Europe. While it is interesting (although not unexpected) that the Japanese population show similarities, the manuscript often implies that these features are being observed for the first time. For example, the fact that many elderly women with breast cancer can be BRCA mutation carriers is not a new idea but the way it is described suggests this is surprising.

“Moreover, even in the patients diagnosed at 80 years or over, the proportion 358 of patients with pathogenic variants was five times higher than that of controls (aged 359 60 and over). These results suggest that pathogenic variants in predisposition genes 360 not only contribute to early-onset breast cancer, but also affect late-onset breast 361 cancer in Japanese.”

In fact, even for BRCA1/2 the penetrance is nowhere near 100%, so clearly there will be elderly mutation carriers. This and other observations need to be conveyed in context of what has been described in previous studies.

6. The authors should make more of their unique control frequency data. If the masses are included, this is a very large cohort and it would be interesting expand on how the frequency compares with European data. In addition there have recently been some smaller studies of Chinese populations. How does the data compare with those?

Reviewer #2:

Remarks to the Author:

This is an extremely well written and timely manuscript. The investigators performed a case control association study for variants in coding regions of 11 known hereditary breast cancer genes using a large data set of female and male breast cancer patients (and controls) from a predominately Asian population. 244 pathogenic variants were identified of which 131 were novel.

The impact of this work lies within the power of the large sample set to validate novel pathogenic variants. The authors have beautifully described the characterization of these variants and have thoughtfully discussed relevance to existing work, predominately in multi-ethnic populations.

They appropriately discuss the limitations of the work and the likely impact of these limitations, which is relatively low.

The authors also attempt to validate the use of NCCN guidelines in a Japanese population. While they do discuss the ability of the guidelines to detect the described pathogenic variants, this piece of the work is somewhat distracting and takes away from the impactful discovery rather than augments it. Additionally, the data set involved did not capture all of the information required to assess patients according to NCCN guidelines, so assumptions were made that may impact accuracy of the work. Would consider removing this information and submitting as a separate manuscript; however, this is not a fatal flaw that would limit publication.

Reviewers' comments:

Reviewer #1 (Remarks to the Author):

This study records the frequency of germline pathogenic mutations in 11 breast cancer predisposition genes among a Japanese population of 7051 women with breast cancers and 11,241 female controls. In addition, 53 male breast cancer patients and 12,520 male controls were studied. Similar studies have been extensively reported on in Western European populations but this is the largest study of a Japanese population. A strength of the study is the large sample size and the inclusion of controls that seem to be reasonable matched to the cases. Technically the study appears to have been conducted well and the variant annotation and pathogenicity calling has been conducted rigorously.

Overall the study provides some insight into the prevalence of breast cancer predisposition gene mutations in an Asian population.

We thank the reviewer for carefully reading our manuscript and for providing useful comments.

There are a number of aspects of the manuscript that limit the impact of the work:

1. The extent to which this data is relevant to other Asian populations where there have been few large-scale studies of this kind is unclear. It would be helpful if the authors could indicate how the Japanese population data might reflect Asian populations in general? How will this work change clinical practice in Japan and elsewhere?

[Response 1] Thank you for indicating these points. As for the impact of this study on the clinical practice in Japan, we identified 244 unique pathogenic variants, of which 131 were novel. According to this comment, we expanded this result to calculate how many samples were newly identified as patients with pathogenic variants. When we used only

information from ClinVar, we identified 258 patients with pathogenic variants and missed 146 patients. Therefore, this study identified 57% (146/258) more patients with pathogenic variants. The following figure shows analyses for each gene separately. More than 75% of patients with pathogenic variants in *BRCA1/2* could be identified by ClinVar only, however, a small proportion of patients with pathogenic variants in other genes, especially *PALB2*, *CHEK2*, *ATM*, and *NF1*, were detected. Therefore, this study contributes to improved identification of patients with pathogenic variants in the diagnosis of hereditary breast cancer in Japan, especially with respect to *PALB2*, *CHEK2*, *ATM*, and *NF1*.

As for the relevance of this study to other Asian population, we refer to Chinese (Int J Cancer 2017, 141, 129–142) and Malaysian (J Med Genet 2018, 55, 97–103) studies. Both studies sequenced coding regions in *BRCA1/2* in >2,000 selected and unselected breast cancer patients, respectively. The Chinese study identified 175 unique

pathogenic variants in 247 of 2,991 (8.3%) patients and 15 of 175 (8.6%) pathogenic variants were shared with our study. Similarly, the Malaysian study identified 97 unique pathogenic variants in 121 of 2,575 (4.7%) patients. Of these pathogenic variants, 15 (15.5%) variants were identified in our study. These results suggest that pathogenic variants identified in this study were shared in Asian populations to some extent. Therefore, this study contributes to the identification of patients with pathogenic variants in the diagnosis of hereditary breast cancer in other Asian countries. However, it will still be necessary to create a list of pathogenic variants based on a large number of samples for improved diagnosis of hereditary breast cancer.

[Add the following description into the Discussion at L. 328-351]

We identified 113 variants previously noted as pathogenic. Data from this study helped to classify 131 additional variants, resulting in a total of 244 pathogenic variants identified. This increase resulted in the identification of 57% more patients (from 258 to 404) with a pathogenic variant. Supplemental Figure 5 shows this change in each gene. Although more than 75% of patients could be identified by only ClinVar in *BRCA1/2*, only a small proportion of patients with other genes, especially *PALB2* (18%), *CHEK2* (8%), *ATM* (24%), and *NF1* (25%), were identified. Therefore, this study contributes to improved identification of patients with a pathogenic variant, especially in genes other than *BRCA1/2*, in the diagnosis of hereditary breast cancer in clinical practice in Japan. Next we investigated the proportion of pathogenic variants shared between other Asian countries and this study to address how the Japanese data are relevant to other Asian populations. Two studies from China³³ and Malaysia³⁴ sequenced *BRCA1/2* in >2,000 selected and unselected breast cancer patients, respectively.

The Chinese study identified 175 unique pathogenic variants in 247 of 2,991 (8.3%) patients. Of the 175 pathogenic variants, 15 (8.6%) pathogenic variants were identified in this study. Similarly, the Malaysian study identified 97 unique pathogenic variants in 121 of 2,575 (4.7%) patients. Of these pathogenic variants, 15 (15.5%) variants were identified in our study. These results suggest that pathogenic variants identified in this study were shared in Asian populations to some extent. Therefore, this study contributes to the identification of patients with a pathogenic variant in the diagnosis of hereditary breast cancer in other Asian countries. However, it will still be necessary to create a list of pathogenic variants based on a large number of samples for improved diagnosis of hereditary breast cancer.

2. The male breast cancer data is based on small numbers but it is not unreasonable to include these in the study. However I am confused as to why the male controls are not included in the overall analysis. While there might be expected to be some minor differences in the frequency of pathogenic mutations in males because there will be no attrition of carriers with breast and ovarian cancer, surely this would be relatively minor?

[Response 2] Thank you for the understanding of the importance of male breast cancer data in this study. The reason why we performed the analysis between men and women separately is that the genetic risk of hereditary breast cancer is known to be different by sex. For example, a previous study that examined 715 male breast cancer patients using multi-gene panel (Breast Cancer Res Tr 2017, 161, 575–586) showed that *BRCA2* and *CHEK2* were the most frequently mutated genes, whereas *BRCA1* was a

low-risk gene (OR = 1.8). Thus, the overall analysis by combining men and women will distort the risk of each variant/gene. Therefore, we performed the analysis by sex throughout the manuscript. We have added the following sentence into the Materials and Method.

[Add the following description into the Methods at L. 102-104]

We analyzed women and men separately, as genetic risk for hereditary breast cancer genes differs between men and women¹².

3. From my perspective the aspect of the study that raises it above the numerous similar studies in European populations is how the frequency of mutations and carrier characteristics differ in Japan and to what extent the lessons learned from western populations will be applicable in Japan and other Asian populations. While these aspects are noted in the discussion, much of the manuscript is devoted to a description of the variant classification, VUS, number of novel variants, distribution of variants across each of the genes and relationship with gene length etc. While this data might should be included as supplementary data, too much of the results and discussion are devoted to these aspects. Overall the manuscript could be shortened considerable by focussing on the novel aspects.

[Response 3] We agree that the description of the variants in the Results section was too extensive. We have moved the following description from the Results section to the Supplemental Note:

[Move the following description to the Supplemental Note at L. 20-29]

Sequencing of the 11 established hereditary breast cancer genes

identified 1,781 germline variants among 7,051 breast cancer cases and 11,241 controls. According to the genomic position, we categorized the variants into 210 disruptive, 1,084 nonsynonymous, and 487 synonymous variants (Supplemental Table 10). Minor allele frequencies (MAF) of these variants in controls were common ($MAF \geq 5\%$) for 30 variants, low ($5\% > MAF \geq 1\%$) for 27 variants, and rare ($MAF < 1\%$) for 1,724 variants. More than half of the variants (all rare) were not registered at dbSNP147¹. When we examined the density of variants in each gene, the number of variants was strongly correlated with the gene length ($r = 0.953$, $p = 5.70 \times 10^{-6}$, Supplemental Figure 6).

4. On page 5 the authors state that they studied 11 established hereditary breast cancer genes but in presenting the data the authors state that only some of these reached significance. Given that most of these genes have very well established roles in breast cancer predisposition, why are those other gene de-emphasised. Particularly for STK11, TP53 and CDH1 where the number of carries is small, these will never reach significance but they are well established cancer predisposition genes. On the other hand, genes such as NBN are not well support in the literature and the data presented here shows only one carrier in the cases and 3 in the controls. This result should be noted and compared with previous studies which I believe also show no association breast cancer predisposition.

[Response 4] Thank you for the suggestion to highlight the importance of results for additional genes in the Discussion section. We have now added more text to the discussion as follows:

[Add the following description into the Discussion at L. 279-310]

The 11 genes analyzed in this study have been reported previously as hereditary breast cancer genes, but the strength of evidence for association of each gene with breast cancer and disease risk varies. Further, published risk estimates are likely to be inflated for at least some genes due to ascertainment bias³. We observed a significant contribution to breast cancer risk in *BRCA1/2*, *PALB2*, and *TP53*. The disease risks of *BRCA1/2* and *PALB2* are comparable to that previously reported³, but the risk of *TP53* is largely different (8.5 in this study and 105 in the previous meta-analysis³). This is likely explained by several factors. Firstly, previous estimates were based on studies of “familial” patients presenting with clinical features of Li–Fraumeni syndrome, whereas in this study we calculated disease risk for women unselected for family history of cancer. Second, functional effects differ between variants in *TP53*, which causes a wide range of symptoms, from the severe form known as Li–Fraumeni to the less severe nonsyndromic predisposition²⁶, and it is possible that the variants found in patients with unselected breast cancer have less impact on protein function than those identified in patients with classical Li–Fraumeni syndrome. Among four other genes showing $P < 0.05$ (*PTEN*, *CHEK2*, *NF1*, and *ATM*), the disease risks of *ATM* and *CHEK2* were comparable to previous reports³. Disease risks for *PTEN* and *NF1* were not reliably estimated, despite strong evidence for association ($P < 5 \times 10^{-4}$), due the low numbers of carriers, indicating need for even larger studies to estimate risk at the population level. Although the association with breast cancer for *CDH1* and *STK11* has been reported previously for patients for hereditary diffuse gastric cancer²⁷ and Peutz–Jeghers syndrome²⁸, only two and zero Japanese breast cancer patients, respectively,

had a pathogenic variant in these genes. That is, *CDH1* and *STK11* have a limited contribution to breast cancer in unselected Japanese women. The reported contribution of *NBN* to breast cancer risk was mainly based on one specific variant (c.657del5, rs587776650) in the Slavic population^{3, 29}, which was not observed in the Japanese population. Other *NBN* variants designated as pathogenic using ACMG criteria were observed in only 1 case and 3 controls, providing little support for a role of *NBN* in Japanese unselected breast cancer patients. However, our study has confirmed the importance of the remaining eight genes in genetic testing in Japan and jointly assessed the disease risk of each gene.

5. Some of the comments in the manuscript describe features of breast cancer predisposition that have been established for many years in European. While it is interesting (although not unexpected) that the Japanese population show similarities, the manuscript often implies that these features are being observed for the first time. For example, the fact that many elderly women with breast cancer can be BRCA mutation carriers is not a new idea but the way it is described suggests this is surprising.

“Moreover, even in the patients diagnosed at 80 years or over, the proportion of patients with pathogenic variants was five times higher than that of controls (aged 60 and over). These results suggest that pathogenic variants in predisposition genes not only contribute to early-onset breast cancer, but also affect late-onset breast cancer in Japanese.”

In fact, even for BRCA1/2 the penetrance is nowhere near 100%, so clearly there

will be elderly mutation carriers. This and other observations need to be conveyed in context of what has been described in previous studies.

[Response 5] We totally agree this comment. We have removed the description related to this comment from the Discussion section.

6. The authors should make more of their unique control frequency data. If the masses are included, this is a very large cohort and it would be interesting expand on how the frequency compares with European data. In addition there have recently been some smaller studies of Chinese populations. How does the data compare with those?

[Response 6] Thank you for the suggestion to compare the control frequency data with European and Chinese populations. Such a comparison would be especially important for rare variants because all pathogenic variants in this study were rare. In this study, we identified 1,724 rare variants with MAF <0.01, and 1,011 (58.6%) were polymorphic in controls. The remaining 713 variants were identified only in cases. In the non-Finnish European (NFE) population in ExAC, only 31 (1.8%) rare variants were found. Frequency information in controls is indispensable for assigning the clinical significance of each variant at PS4 of the ACMG/AMP guidelines and for estimating disease risk. However, almost none of the rare variants were found in ExAC. This suggests that NFE in ExAC might not be suitable as an alternative for Japanese control samples.

This would be the same in the East Asian population. The Chinese study to which you are likely referring (Int J Cancer 2017, 141, 129–142) did not provide frequency information for any of the variants in controls. Instead, we used the East Asian (EAS) population in ExAC to compare the frequency with our study. As in the comparison with NFE, most rare variants (95.0%) were not found in EAS. These results

suggest the importance of large-scale population-matched controls for assigning clinical significance and estimating disease risk.

According to this comment, we have added the following description to the Supplemental Note:

[Add the following description into the Supplemental Note at L. 127-142]

3. Comparison of variant frequency in controls.

This study analyzed 11,241 female controls, but other studies have used data from the Exome Aggregation Consortium (ExAC)⁸ as a control for the estimation of disease risk¹⁹. We investigated the difference in allele frequency between Japanese women in this study and East Asian (EAS) and non-Finnish European (NFE) populations from ExAC without the Cancer Genome Atlas samples. We focused on rare variants with MAF <0.01 because all pathogenic variants were rare. In this study, we identified 1,724 rare variants, of which 1,011 (58.6%) were polymorphic in the controls and the remaining 713 variants were identified only in cases. However, only 87 (5.0%) and 31 (1.8%) were found in the EAS and NFE populations of ExAC, respectively. The frequency of relevant controls is indispensable for assigning clinical significance at PS4 of the ACMG/AMP guidelines and for estimating disease risk. However, because most rare variants were not found in ExAC, population-matched controls are necessary for appropriate assignment of clinical significance and better estimation of disease risk.

Reviewer #2 (Remarks to the Author):

This is an extremely well written and timely manuscript. The investigators performed a case control association study for variants in coding regions of 11 known hereditary breast cancer genes using a large data set of female and male breast cancer patients (and controls) from a predominately Asian population. 244 pathogenic variants were identified of which 131 were novel.

The impact of this work lies within the power of the large sample set to validate novel pathogenic variants. The authors have beautifully described the characterization of these variants and have thoughtfully discussed relevance to existing work, predominately in multi-ethnic populations.

They appropriately discuss the limitations of the work and the likely impact of these limitations, which is relatively low.

We thank the reviewer for carefully reading our manuscript and for providing useful comments.

The authors also attempt to validate the use of NCCN guidelines in a Japanese population. While they do discuss the ability of the guidelines to detect the described pathogenic variants, this piece of the work is somewhat distracting and takes away from the impactful discovery rather than augments it. Additionally, the data set involved did not capture all of the information required to assess patients according to NCCN guidelines, so assumptions were made that may impact accuracy of the work. Would consider removing this information and submitting as a separate manuscript; however, this is not a fatal flaw that would limit publication.

[Response] Thank you for raising the important point. As the reviewer indicated, we could not include all information required by the NCCN guidelines because clinical data

of Biobank Japan were not collected for the aim of hereditary breast cancer. We agree that this distracts the impact of our discovery and thus have removed all related description from the main text.

However, we compared the proportion of pathogenic variants in each gene between this study and the largest study conducted with 35,409 multiethnic women (Cancer 2017, 123, 1721–1730) in the Discussion section. To do this comparison, it was necessary to select patients using the NCCN guidelines, as was done in the large study noted previously. Therefore, we have kept the description of the NCCN guidelines for this comparison in the Supplemental Note. Because the purpose of describing the NCCN guidelines changed from assessment of the guidelines to selection of patients according to the guidelines, we have modified the description as follows:

[Add the following description into the Supplemental Note at L. 104-125]

2. Selection of patients according to NCCN guidelines

We selected patients by the National Comprehensive Cancer Network (NCCN) guidelines for genetic/familial high-risk assessment of breast and/or ovarian cancer (ver. 2.2016)¹⁷ for the comparison of proportion of patients with a pathogenic variant with another study¹⁸ because they selected patients based on the NCCN guidelines. Since Biobank Japan did not collect clinical information of breast cancer as hereditary disease, we did not have some information for family members (a known mutation of hereditary breast cancer genes within the family, and age at diagnosis of breast cancer and histology of ovarian cancer in close relatives). Thus, we slightly modified the criteria as follows. (1) Age at breast cancer diagnosis \leq 50 years old, (2) triple negative breast cancer diagnosed at \leq 60 years old, (3) bilateral breast cancer, (4) comorbidity of pancreatic cancer at

any age, (5) ≥ 1 family member with breast cancer at any age (not ≤ 50 years), (6) ≥ 1 family member with ovarian cancer (not invasive ovarian cancer) at any age, and (7) ≥ 2 family members with breast and/or pancreatic cancer at any age. Patients who met at least one of the criteria were treated as high-risk for further genetic risk evaluation. Patients who did not meet any of the criteria were considered as low-risk. If patients were not classified either high-risk or low-risk due to insufficient clinical information, they were considered as undetermined-risk. As results, 3,136 patients were high-risk, 1,164 were low-risk and 2,751 were undetermined-risk (Supplemental Table 15). The 3,136 high-risk patients were used for the comparison of proportion of patients with a pathogenic variant.

Reviewers' Comments:

Reviewer #1:

Remarks to the Author:

The authors have provided a very good response to most of the reviewers comments and the amendments made to the manuscripts are appropriate and sufficient. My only remaining issue is the male controls. My reference to using the males was not to combine the male and female CASES but combining the CONTROLS. I agree the pathogenic mutations in the male and female breast cancers may be different but there is no reason to expect that there will be any difference in the population frequency of breast cancer genes among the control groups. I still believe the manuscript would be more powerful if the female and male controls were combined rather than just using the female controls.

Reviewer #2:

Remarks to the Author:

The authors have answered this reviewer's concerns and I would recommend publication.

Reviewer #1 (Remarks to the Author):

The authors have provided a very good response to most of the reviewers comments and the amendments made to the manuscripts are appropriate and sufficient. My only remaining issue is the male controls. My reference to using the males was not to combine the male and female CASES but combining the CONTROLS. I agree the pathogenic mutations in the male and female breast cancers may be different but there is no reason to expect that there will be any difference in the population frequency of breast cancer genes among the control groups. I still believe the manuscript would be more powerful if the female and male controls were combined rather than just using the female controls.

[Response] Thank you very much for appreciating our previous revision and clarifying this point. According to this valuable comment, we combined both female and male controls and determined clinical significance of all variants again. First, we focused 1,781 variants found in women to check how the increased number of controls improved the determination of clinical significance. Table 1 shows the comparison of clinical significance between use of only female controls and use of both controls. We observed that only one variant (p.Leu3048Phe in *ATM*) changed from “uncertain significance” to “pathogenic” because this variant came to meet PS4 of the ACMG guidelines. As a result, the combining female and male controls did not change the pathogenicity of many variants.

Table 1: Comparison of clinical significance in 1,781 variants

		Use of female and male controls		
		Pathogenic	Benign	Uncertain significance
Use of only female controls	Pathogenic	244	0	0
	Benign	0	356	0
	Uncertain significance	1	7	1,173

Then, we performed gene-based analysis with 245 pathogenic variants and 39 additional pathogenic variants found in only male controls in 7,051 cases and 23,731 controls (Table 2). As comparison, original results in 7,051 cases and 11,241 female controls are also shown in Table 3. As a whole, results were very similar between two tables. However, when we checked each gene separately, we observed that odds ratio of *BRCA1* largely decreased from 33.0 to 20.5. The frequency of controls with pathogenic variants increased from 0.04% to 0.07% by adding male controls. Among controls, men had more pathogenic variants in *BRCA1* (0.1%) than women (0.04%). This result is consistent with the recent publication about male breast cancer (*Breast Cancer Res Tr* 2017, 161, 575–586) which showed *BRCA1* was a low-risk gene (OR = 1.8). Therefore, female disease risk of *BRCA1* would be underestimated. These results suggest that combining

female and male controls would introduce bias of disease risk estimation when disease risk of a gene is different between both sexes.

Taken together, while we kept analysis of female and male controls separately, we considered this analysis proposed by the reviewer was very valuable in this field. We added this discussion into the Supplemental Note and Tables 1 and 2 as Supplemental Tables. We cited this Supplemental Note at the Materials and Methods and the Discussion in the main text.

Table 2: Result of gene-based association test using pathogenic variants in 7,051 cases and 23,731 female and male controls.

Gene	No. of pathogenic variants	Case (n = 7,051)	Control (n = 23,731)	P value	OR	(95% CI)
		No. of carriers (%)	No. of carriers (%)			
BRCA2	92	191 (2.71)	44 (0.19)	1.23×10^{-80}	15.0	(10.7-21.3)
BRCA1	57	102 (1.45)	17 (0.07)	6.40×10^{-48}	20.5	(12.2-36.5)
PALB2	23	28 (0.40)	9 (0.04)	1.53×10^{-11}	10.5	(4.8-25.3)
TP53	15	16 (0.23)	6 (0.03)	9.92×10^{-7}	9.0	(3.3-28.1)
CHEK2	18	26 (0.37)	23 (0.10)	7.25×10^{-6}	3.8	(2.1-7.0)
PTEN	15	11 (0.16)	4 (0.02)	4.84×10^{-5}	9.3	(2.7-39.9)
ATM	39	26 (0.37)	39 (0.16)	0.003	2.2	(1.3-3.8)
NF1	13	8 (0.11)	5 (0.02)	0.003	5.4	(1.6-20.9)
CDH1	3	2 (0.03)	1 (0.00)	0.133	6.7	(0.4-396.2)
NBN	7	1 (0.01)	7 (0.03)	0.692	0.5	(0.0-3.7)
STK11	2	0 (0.00)	2 (0.01)	1.000	0.0	(0.0-17.9)
Sum	284	408 (5.79)	157 (0.66)	5.99×10^{-133}	8.7	(7.2-10.6)

(cf.) Table 3: Result of gene-based association test using pathogenic variants in 7,051 cases and 11,241 controls.

Gene	No. of pathogenic variants	Case (n = 7,051)	Control (n = 11,241)	P value	OR	(95% CI)
		No. of carriers (%)	No. of carriers (%)			
BRCA2	85	191 (2.71)	19 (0.17)	9.87×10^{-58}	16.4	(10.2-28.0)
BRCA1	55	102 (1.45)	5 (0.04)	3.71×10^{-36}	33.0	(13.7-103.8)
PALB2	21	28 (0.40)	5 (0.04)	5.79×10^{-8}	9.0	(3.4-29.7)
TP53	13	16 (0.23)	3 (0.03)	5.93×10^{-5}	8.5	(2.4-45.6)
PTEN	12	11 (0.16)	1 (0.01)	2.16×10^{-4}	17.6	(2.6-753.3)
CHEK2	17	26 (0.37)	13 (0.12)	4.31×10^{-4}	3.2	(1.6-6.8)

NF1	8	8 (0.11)	0 (0.00)	4.86 x 10 ⁻⁴	Inf	(2.7-Inf)
ATM	27	22 (0.31)	17 (0.15)	0.031	2.1	(1.0-4.1)
CDH1	2	2 (0.03)	0 (0.00)	0.149	Inf	(0.3-Inf)
NBN	3	1 (0.01)	3 (0.03)	1.000	0.5	(0.0-6.6)
STK11	1	0 (0.00)	1 (0.01)	1.000	0.0	(0.0-62.1)
Sum	244	404* (5.73)	67 (0.60)	2.87 x10 ⁻¹⁰²	10.1	(7.8-13.4)

[Add the following description into Supplemental Note]

Supplemental Note 1: Influence of combining female and male controls

In this study, we analyzed women and men separately, as genetic risk for hereditary breast cancer differs between men and women¹. However, there is a possibility to assign more variants as pathogenic by use of both female and male controls because the number of controls increases twofold from 11,241 to 23,731. To test this possibility, we combined both controls and determined clinical significance of all variants again. First, we focused the 1,781 variants found in women to check how the increased number of controls improved the determination of clinical significance. As in the Supplemental Table 10, we observed that only one variant (p.Leu3048Phe in *ATM*) changed from “uncertain significance” to “pathogenic” because this variant came to meet PS4 of the ACMG guidelines. As a result, the combining female and male controls did not change the pathogenicity of many variants.

Then, we performed gene-based analysis with 245 pathogenic variants found in women and 39 additional pathogenic variants found in only male controls in 7,051 cases and 23,731 female and male controls (Supplemental Table 11). As a whole, results were very similar to Table 2 analyzed in 7,051 cases and 11,241 female controls only. However, when we checked each gene separately, we observed that odds ratio of *BRCA1* largely decreased from 33.0 to 20.5 because the frequency of controls with pathogenic variants increased from 0.04% to 0.07% by adding male controls. Among controls, men had more pathogenic variants in *BRCA1* (0.1%) than women (0.04%). This result is consistent with the recent publication about male breast cancer¹ which showed *BRCA1* was a low-risk gene (OR = 1.8). Therefore, female disease risk of *BRCA1* would be underestimated.

These results suggest that combining male and female controls would introduce bias of disease risk estimation when disease risk of a gene is different between both sexes.

Reviewer #2 (Remarks to the Author):

The authors have answered this reviewer's concerns and I would recommend publication.

Thank you very much for this recommendation.

Reviewers' Comments:

Reviewer #1:

Remarks to the Author:

The authors have provided a good rebuttal regarding the male controls and their modifications to accommodate my suggestion and their worry about bias seems very reasonable.

REVIEWERS' COMMENTS:

Reviewer #1 (Remarks to the Author):

The authors have provided a good rebuttal regarding the male controls and their modifications to accommodate my suggestion and their worry about bias seems very reasonable.

[Response] Thank you very much for understanding our modifications.